# Yellow Emission Obtained by Combination of Broadband Emission and Multi-Peak Emission in Garnet Structure Na_2_YMg_2_V_3_O_12_: Dy^3+^ Phosphor

**DOI:** 10.3390/molecules25030542

**Published:** 2020-01-27

**Authors:** Weiyi Zhang, Can He, Xiaowen Wu, Ximing Huang, Fankai Lin, Yan’gai Liu, Minghao Fang, Xin Min, Zhaohui Huang

**Affiliations:** Beijing Key Laboratory of Materials Utilization of Nonmetallic Minerals and Solid Wastes, National Laboratory of Mineral Materials, School of Materials Science and Technology, China University of Geosciences, Beijing 100083, China; zhangweiyi1993@163.com (W.Z.); hecan1211@163.com (C.H.); xmhuang1995@163.com (X.H.); linfankai85@163.com (F.L.); liuyang@cugb.edu.cn (Y.L.); fmh@cugb.edu.cn (M.F.); minxin4686@126.com (X.M.)

**Keywords:** Na_2_YMg_2_V_3_O_12_:Dy^3+^, yellow emitting, luminescence, WLED

## Abstract

The fabrication and luminescent performance of novel phosphors Na_2_YMg_2_V_3_O_12_:Dy^3+^ were investigated by a conventional solid-state reaction method. Under near-UV light, the Na_2_YMg_2_V_3_O_12_ host self-activated and released a broad emission band (400–700 nm, with a peak at 524 nm) ascribable to charge transfer in the (VO_4_)^3−^ groups. Meanwhile, the Na_2_YMg_2_V_3_O_12_:Dy^3+^ phosphors emitted bright yellow light within both the broad emission band of the (VO_4_)^3-^ groups and the sharp peaks of the Dy^3+^ ions at 490, 582, and 663 nm at a quenching concentration of 0.03 mol. The emission of the as-prepared Na_2_YMg_2_V_3_O_12_:Dy^3+^ phosphors remained stable at high temperatures. The obtained phosphors, commercial Y_2_O_3_:Eu^3+^ red phosphors, and BaMgAl_10_O_17_:Eu^2+^ blue phosphors were packed into a white light-emitting diode (WLED) device with a near-UV chip. The designed WLED emitted bright white light with good chromaticity coordinates (0.331, 0.361), satisfactory color rendering index (80.2), and proper correlation to a color temperature (7364 K). These results indicate the potential utility of Na_2_YMg_2_V_3_O_12_:Dy^3+^ phosphor as a yellow-emitting phosphor in solid-state illumination.

## 1. Introduction

For several years, phosphor-converted white light-emitting diodes (pc-WLEDs) have been recognized as the most promising replacements of traditional incandescent and fluorescence lamps [1,2,3,4,5]. These solid-state light sources deliver high luminescence efficiency and an excellent operation lifetime (>10,000 h) while conserving energy and lowering the pollution risk. Most commercial pc-WLEDs are obtained by two methods [6,7]. One method generates white light by combining yellow phosphors with blue LED chips, such as the yellow phosphor YAG: Ce^3+^ [8,9]. However, the absence of the red-emitting component reduces the quality of the white light, yielding poor color reproduction and a low color rendering index (R_a_) [10]. The other method stimulates blue, green, and red (RGB) phosphors by violet or ultraviolet light LEDs [11]. Although this method improves the R_a_ and can tune the correlated color temperature (CCT), the emission efficiency is limited by reabsorption among the different phosphors [1,12]. The different thermal stabilities and ageing rates of the various phosphors also restrict their applications in WLEDs [13]. Therefore, high-performance single-phased phosphors that can be excited by ultraviolet (UV) or near-UV (n-UV) light are urgently needed [14,15,16]. A single-host white-emitting phosphor must usually have a broad emission peak or multiple emission peaks in the visible region. Therefore, searching for new broadband emission or multi-peak emission phosphors is significant for improving the color stability and service life of WLEDs excited by UV or n-UV light [17,18].

In recent decades, the rare earth luminescent materials have been used in many fields, such as lighting, photosynthesis enhancement, and photodynamic activation for cancer treatments [19,20,21]. Broad emission peaks or multiple emission peaks in the visible region facilitate white light emission with appropriate R_a_ and CCT values [22,23,24]. Vanadate composed of VO_4_^3−^ tetrahedrons is an important non-rare earth ion-doped luminescent material displaying broadband emission, excellent luminous efficiency, and good chemical stability [25,26]. For example, A_3_(VO_4_)_2_ (A = Mg, Sr, Ba) phosphors yield intense charge transfer (CT) absorption bands in the n-UV region and broad emission bands covering 400–700 nm. Further studies have reported that when doped with rare earth ions, vanadate is a good host material that enhances the emission efficiency of phosphors [27,28]. Guo et al. reported that Eu^3+^-activated Ba_2_BiV_3_O_11_ phosphors are promising candidates for red-emitting phosphors in WLEDs, as they efficiently convert UV light from 394 nm onwards into red light [29]. Bright orange-red emission has been obtained by doping Sm^3+^ in NaSrVO_4_ phosphor under n-UV light excitation [30]. Song et al. [17] studied self-activated Na_2_YMg_2_V_3_O_12_ vanadate phosphors, and reported a broad emission band of 400–700 nm centered at 520 nm. However, the red emission intensity was relatively low, below the requirements of white light emission. Dy^3+^ doping should broaden the emission band of Na_2_YMg_2_V_3_O_12_ phosphors.

In this work, a series of yellow-emitting Na_2_YMg_2_V_3_O_12_:Dy^3+^ phosphors was prepared by a conventional high-temperature solid-state method. The phase purities, micromorphologies, luminescence properties, and decay times of the as-prepared Na_2_YMg_2_V_3_O_12_:Dy^3+^ phosphors were studied in detail. The suitability of the yellow-emitting phosphors for indoor illumination was demonstrated in a WLED device incorporating the developed phosphors.

## 2. Results and Discussion

The phase compositions and crystal structures of the as-prepared powder samples were characterized at room temperature. The XRD patterns of Na_2_YMg_2_V_3_O_12_:*x*Dy^3+^ (*x* = 0, 0.005, 0.01, 0.03, 0.05, 0.07) samples exhibited main peaks at 17.5°, 20.3°, 28.8°, 32.3°, 33.9°, 35.5°, 36.9°, 51.0°, 53.2°, and 55.4° (Figure 1a), corresponding, respectively, to the (2 1 1), (2 2 0), (4 0 0), (4 2 0), (3 3 2), (4 2 2), (4 3 1), (4 4 4), (6 4 0), and (6 4 2) facets of a single garnet structure with a cubic Ia3d (No. 230) space group. All diffraction peaks of the Na_2_YMg_2_V_3_O_12_:Dy^3+^ samples were well matched with the standard profile (PDF No.49-0412), confirming that doping with Y^3+^ ions did not significantly affect the crystalline structure of Na_2_YMg_2_V_3_O_12_.

Figure 1b shows the spatial structure of the unit cell of the garnet-structured Na_2_YMg_2_V_3_O_12_. As implied, the A sites were occupied by alkaline metal ions Na^+^ and rare earth ions Y^3+^, which were coordinated with eight oxygen O^2−^ ions to form a dodecahedron with D_2_ symmetry (without an inverse center). The alkaline-earth metal Mg^2+^ ions located in the octahedral sites B bonded with six oxygen atoms, and the metal ion V^5+^ (in VO_4_^3−^) occupied the T_d_ sites and were surrounded by four O^2−^ ions. As Y^3+^ and Dy^3+^ have similar cationic radii and the same valence, the Y^3+^ ions in the host lattice were easily replaced by Dy^3+^ ions with no structural transformation. The XRD patterns of Na_2_YMg_2_V_3_O_12_:Dy^3+^ match those of the standard card, further confirming that the Dy^3+^ ions doped in the Na_2_YMg_2_V_3_O_12_ host had replaced the Y^3+^ sites.

The microscopic morphology, particle size, and grain shape of a phosphor are important factors in applications. Field emission scanning electron microscopy (FESEM) images of the Na_2_YMg_2_V_3_O_12_:0.03Dy^3+^ sample confirmed that all particles were irregular oblate spheres with an average particle size of 1 μm (Figure 1c). The spherical morphology was similar to that of commercial YAG: Ce^3+^ phosphor, which possesses the same garnet structure. The average particle size of the prepared phosphor was also similar to that of commercial phosphors. This size may enhance the dispersion and transparency of phosphors in the glue when packaging with the WLEDs.

The photoluminescence (PL) and PL emission (PLE) spectra of the undoped Na_2_YMg_2_V_3_O_12_ sample are presented in Figure 2a–c. Na_2_YMg_2_V_3_O_12_ shows a broad absorption band of 250–400 nm, matching the absorption of near-UV chips in WLEDs. When excited at 289 nm and 365 nm, the as-prepared particles also emitted a broad emission band, ranging from 400 to 700 nm with a maximum at 524 nm. This emission was attributed to the CT of an electron from the 2p orbital of oxygen to the vacant 3d orbital of V^5+^ in the tetrahedral (VO_4_)^3−^ groups [31,32]. The emission band centered at 524 nm was decomposed into two sub-bands by Gaussian peak separation, one centered at 289 nm (4.30 eV), the other at 365 nm (3.41 eV) [33]. As shown in Figure 2d, the (VO_4_)^3−^ group has a ground state ^1^A_1_ and excited states ^1^T_1_, ^1^T_2_, ^3^T_1_, and ^3^T_2_. The decomposed emission sub-bands were attributed to ^3^T_2_→^1^A_1_ (Em1 = 512 nm (2.43 eV)) and ^3^T_1_→^1^A_1_ (Em2 = 571 nm (2.18 eV)) transitions of the (VO_4_)^3−^ groups, respectively. The excitation band was also composed of two sub-bands, which were assigned to the ^1^A_1_→^1^T_2_ (Ex1 = 4.30 eV) and ^1^A_1_→^1^T_1_ (Ex2 = 3.41 eV) transitions of the (VO_4_)^3−^ groups.

Figure 3a shows the PLE (*λ*_em_ = 582 nm) and PL (*λ*_ex_ = 289 and 365 nm) spectra of the Na_2_YMg_2_V_3_O_12_:0.03Dy^3+^ phosphors at room temperature. The broad emission band at 524 nm was assigned to the CT transitions of the (VO_4_)^3−^ groups, and the emission peaks at 490, 582, and 663 nm were, respectively, attributed to ^4^F_9/2_→^6^H_15/2_, ^4^F_9/2_→^6^H_13/2_, and ^4^F_9/2_→^6^H_11/2_ transitions of Dy^3+^. Under excitation at 289 nm and 365 nm, the intensity ratios of the (VO_4_)^3−^ and Dy^3+^ emissions changed because the excitation pathways of Dy^3+^ luminescence depend on the excitation wavelength. When excited at 289 nm and 365 nm, the Dy^3+^ emission was mainly caused by Dy-O CT and by energy transfer from the absorption of V-O CT, respectively [12,34]. Monitoring the phosphor emission under 582 nm, the broad excitation band from 250 to 400 nm (which peaks at two sites: 289 nm and 365 nm) resembles the excitation spectrum of non-doped Na_2_YMg_2_V_3_O_12_. This may have resulted from the energy transfer behavior from Na_2_YMg_2_V_3_O_12_ to Dy^3+^ ions, which completely overlaps the excitation spectrum of Na_2_YMg_2_V_3_O_12_ to Dy^3+^. The broad excitation spectrum indicates that the Na_2_YMg_2_V_3_O_12_:Dy^3+^ sample can be efficiently excited under n-UV light, and can be well matched with n-UV LED chips.

The PL spectra of Na_2_YMg_2_V_3_O_12_:*x*Dy^3+^ (*x* = 0, 0.005, 0.01, 0.03, 0.05, 0.07) samples with different doping concentrations are shown in Figure 3b,c. As the Dy^3+^ concentration increased, the intensities of the emission peaks increased to a maximum at *x* = 0.03, and then decreased under the concentration quenching effect [35]. To investigate the cause of concentration quenching, the interaction type between two excitations was calculated by the following formula:(1)$$\frac{I}{x}=\frac{k}{1+\beta x^{Q/3}}$$
where *k* and *β* are constants, *I* is the emission intensity, and *Q* represents the interaction type. When *Q* is 3, 6, 8, and 10, the interactions are exchange, dipole–dipole, dipole–quadrupole, and quadrupole–quadrupole interactions, respectively. The *Q* value was obtained by linear fitting of the relationship between lg(*I*/*x*) and lg(*x*). When the phosphors were excited at 289 and 365 nm, the slopes (−*Q*/3) were determined as −0.995 and −0.968, respectively (Figure 3d). Both *Q* values were close to the theoretical value of 3.0, indicating that at higher concentrations, the intensity of the Na_2_YMg_2_V_3_O_12_:Dy^3+^ phosphors was quenched by exchange interactions. The excitation spectra of Na_2_YMg_2_V_3_O_12_:xDy^3+^ monitored at 582 nm were also optimized at *x* = 0.03 (Figure 3e). The CIE (International Commission on illumination) chromaticity coordinates of the Na_2_YMg_2_V_3_O_12_:0.03Dy^3+^ sample were determined as (0.357, 0.461) (Figure 3f). The yellow light emission was the combination of the self-activated emission of the Na_2_YMg_2_V_3_O_12_ host with the dominant 4f–4f transitions of the Dy^3+^ ion [36].

To understand the behaviors of the synthesized compounds, the Na_2_YMg_2_V_3_O_12_:xDy^3+^ phosphors were excited at 289 and 365 nm, and their PL decay curves were recorded at 582 nm. The results are shown in Figure 4. The decay curves of Na_2_YMg_2_V_3_O_12_:xDy^3+^ were well fitted to the following exponential function [37]:(2)$$I_{t}=Ae^{-\frac{t}{\tau }}+I_{0}$$
where *I_t_* and *I*_0_ are the emission intensities at time *t* and the initial time, respectively, and *A* is a constant. *τ* determines the decay time. The average lifetimes of the Na_2_YMg_2_V_3_O_12_:*x*Dy^3+^ phosphors with *x* = 0, 0.005, 0.01, 0.03, 0.05, and 0.07 were determined as 1.60, 1.58, 1.54, 1.48, 1.42, and 1.44 μs, respectively, at *λ*_ex_ = 289 nm, and as 1.53, 1.52, 1.49, 1.50, 1.41, and 1.40 μs, respectively, at *λ*_ex_ =365 nm. The PL lifetimes of the Na_2_YMg_2_V_3_O_12_:*x*Dy^3+^ were similar under both excitation wavelengths, possibly reflecting the similar energy transfer behaviors between the vanadate host and Dy^3+^.

High thermal resistance of phosphors is very important for practical applications in solid-state lighting, as it ensures high optical performance of the WLED device. The thermal quenching performance of Na_2_YMg_2_V_3_O_12_:0.03Dy^3+^ phosphor was assessed from the temperature-dependent emission spectra under excitation at 289 and 365 nm. As shown in Figure 5a,c, the emission intensity reduced smoothly as the temperature increased, because the probability of non-radiative transitions increases at higher temperatures. As shown in the insets of Figure 5a,c, the PL integral intensities at 100 °C were 61.6% (*λ*_ex_ = 289 nm) and 61.48% (*λ*_ex_ = 365 nm) of their room temperature intensities. However, the emission positions in the temperature-dependent emission spectra were relatively robust to temperature changes.

To further investigate the thermal stability of this phosphor, the activation energy (Δ*E*) of Na_2_YMg_2_V_3_O_12_:0.03Dy^3+^ was calculated by the Arrhenius equation [38]:(3)$$I\left(T\right)=\frac{I_{0}}{1+ce^{-\frac{\Delta E}{kT}}}$$
where *I*_0_ is the emission intensity of the phosphor at room temperature, *I*(*T*) is the temperature-dependent intensity, *c* is a constant, and *k* is the Boltzmann constant (8.629 × 10^−5^ eV K^−1^). From the slopes of the ln[*I*_0_/*I*(T) − 1] versus 1/*kT* plots (Figure 5b,d), which were well fitted to Equation (3), the Δ*E*s were determined as 0.21 and 0.26 eV under excitation at 289 and 365 nm, respectively. Table 1 compares the CIE chromaticity coordinates, CCT and lifetimes of Na_2_YMg_2_V_3_O_12_:0.03Dy^3+^, and other Dy^3+^-doped phosphors [39,40,41]. The obtained Na_2_YMg_2_V_3_O_12_:Dy^3+^ phosphors presented relatively high thermal stability and are potentially applicable to WLEDs.

To further prove the feasibility of the as-prepared phosphors in solid-state illumination, we designed and packaged WLED devices based on an n-UV chip (365 nm) and the Na_2_YMg_2_V_3_O_12_:Dy^3+^ phosphors. To compensate for the color combination imbalance and improve the R_a_ of the LEDs, we added small amounts of commercial Y_2_O_3_:Eu^3+^ red phosphors and BaMgAl_10_O_17_:Eu^2+^ blue phosphors, thereby fabricating a warm white-emitting LED. Figure 6 shows the electroluminescence spectra and photographs of the as-fabricated LED devices. Obviously, after adding the red and blue phosphors, the emission light of the LED device changed from yellow to white. The CIE coordinates, R_a_ value, and CCT of the white light generated from the LED device (Figure 6c) were (0.331, 0.361), 80.2, and 7364 K, respectively. The CIE chromaticity coordinates of the LED device are also given in Figure 7. The fabricated device yielded a warm white light. The results demonstrate that the as-prepared phosphors are promising yellow-emitting phosphors for indoor solid-state illumination.

## 3. Materials and Methods

The Na_2_YMg_2_V_3_O_12_:xDy^3+^ (x = 0, 0.005, 0.01, 0.03, 0.05, 0.07) phosphors were prepared through a solid state reaction method. The analytical reagent Mg(OH)_2_ (average particle size, d_50_~3.798 μm), NaHCO_3_ (d_50_~5.638 μm), NH_4_VO_3_ (d_50_~160.3 μm), and high pure rare earth oxides Y_2_O_3_ (99.99%, d_50_~ 3.869 μm) and Dy_2_O_3_ (99.9% d_50_~3.990 μm) were used as raw materials. It is noteworthy that 5 mol% excess NaHCO_3_ was needed to compensate for the volatilization loss. The raw materials were mixed thoroughly in agate mortar for 30 min and then put into a crucible with a lid. These mixed chemicals were preheated in a muffle furnace at 500 °C for 6 h, and then heated at 800 °C for 6 h in air. After cooling to room temperature naturally, the samples were ground into powders for measurement.

The LED devices were fabricated with the as-prepared Na_2_YMg_2_V_3_O_12_:0.03Dy^3+^ phosphor, commercial Y_2_O_3_:Eu^3+^ red phosphors, BaMgAl_10_O_17_:Eu^2+^ blue phosphors, and an InGaN chip with a dominant emission at 365 nm (Shenzhen Looking Long Technology Co., Shenzhen, China). First, the phosphors were thoroughly mixed with organic silica gel. The weight ratio of total phosphors to organic silica gel is about 1:5. The silica gel used to package LED chips needs excellent light transmittance [46], and the light transmittance of the silica gel we chose is 96% (thickness of 1mm). Then, the surfaces of the InGaN chips were coated with the mixture with an approximate thickness of 0.5 mm. Finally, the chips were dried at 135 °C for 2h and the LED devices was obtained.

The powder X-ray diffractometer (XRD-6000, SHIMADZU, Kyoto, Japan) with Cu Kα radiation (λ = 0.15406 nm) was used to measure the phase composition of the as-prepared samples with a 40 kV operating voltage and 30 mA current. The microscopic morphology of the as-synthesized sample was investigated by a field-emission scanning electron microscope (SEM, Model Zeiss Supra-55, Heidenheim, Germany). The fluorescence spectrophotometer (F-4600, HITACHI, Tokyo, Japan) equipped with a 150 W Xe lamp as an excitation source was utilized to measure the photoluminescence (PL) and photoluminescence excitation (PLE) spectra at room temperature under 400 V of operating voltage (Xe lamp). The photoluminescence spectrum of the selected phosphor, which is temperature-dependent, was examined using a computer-controlled electric furnace spectrophotometer (TAP02, Orient KOJI, Tianjin, China). The phosphor powder was heated with a heating rate of 50 °C/min, and held at each test temperature for 3 min. The PL decay curves were obtained through a spectrofluorometer (TBX-PS; HORIBA Jobin Yvon, Paris, France) monitoring at 593nm under excitations of 289 nm and 365 nm, respectively. The electroluminescence spectra, CCT, and Ra of the packed LED devices were measured using a UV-vis-near IR spectrophotocolorimeter (PMS-80, Everfine, Hangzhou, China). 

## 4. Conclusions

In summary, a series of vanadate phosphors Na_2_YMg_2_V_3_O_12_:Dy^3+^ was synthesized by the conventional solid-state reaction method at 800 °C for 6 h. In the XRD analysis, the as-prepared phosphors were found to crystallize in a single garnet structure with a cubic Ia3d (230) space group. When excited by near-UV light, the Na_2_YMg_2_V_3_O_12_ host was self-activated and emitted a broad emission band of 400–700 nm with a peak at 524 nm. This emission was ascribed to CT in the (VO_4_)^3−^ groups. Meanwhile, the Na_2_YMg_2_V_3_O_12_:Dy^3+^ phosphors showed both the broadband luminescence of the (VO_4_)^3−^ groups and the sharp peak emissions of Dy^3+^ ions, and emitted intense yellow light. The phosphors were also excited by light at 289 and 365 nm, and the optimum Dy^3+^ concentration was around 0.03 mol. The temperature-dependent emission spectra indicated high thermal stability of the Na_2_YMg_2_V_3_O_12_:Dy^3+^ phosphors. Finally, a WLED device based on n-UV chip, Na_2_YMg_2_V_3_O_12_:0.03Dy^3+^, Y_2_O_3_:Eu^3+^, and BaMgAl_10_O_17_:Eu^2+^ presented an intense white light with CIE coordinates, color rendering index, and CCT of (0.331, 0.361), 80.2, and 7364 K, respectively. These results suggest the suitability of Na_2_YMg_2_V_3_O_12_:Dy^3+^ phosphor as a yellow-emitting phosphor in WLEDs.

## Figures and Tables

**Figure 1 molecules-25-00542-f001:**
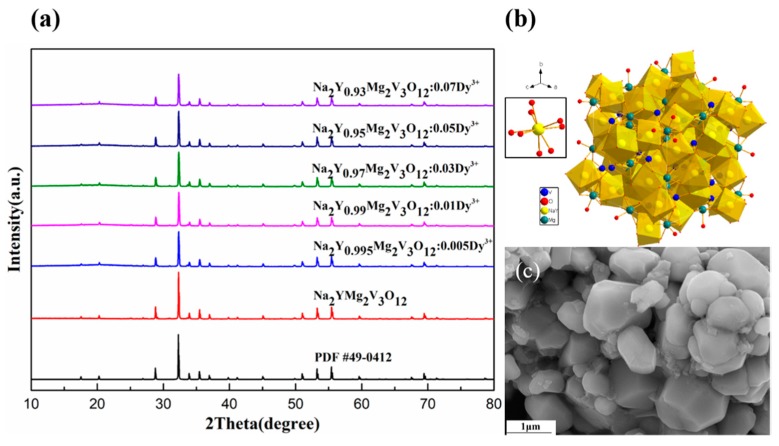
(**a**) XRD patterns of Na_2_YMg_2_V_3_O_12_:Dy^3+^ phosphors and the standard profile (Pdf NO. 49-0412), (**b**) schematic of the crystal structure of Na_2_YMg_2_V_3_O_12_, and (**c**) FESEM micrograph of the Na_2_YMg_2_V_3_O_12_:0.03Dy^3+^ phosphor.

**Figure 2 molecules-25-00542-f002:**
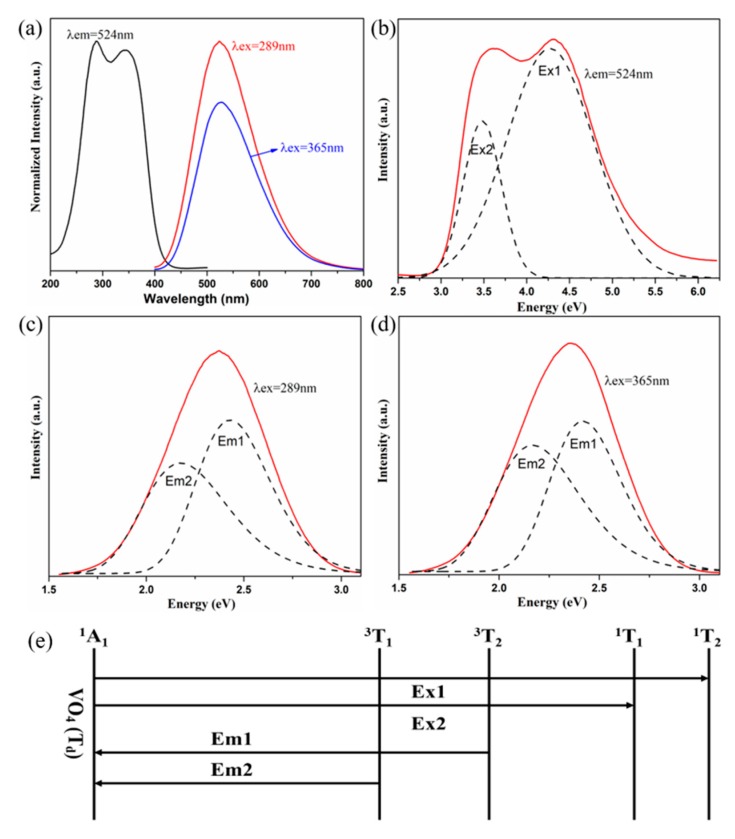
(**a**) Photoluminescence (PL) and PL emission (PLE) spectra of the Na_2_YMg_2_V_3_O_12_ samples, (**b**), (**c**), and (**d**) PLE and PL spectra of the samples after Gaussian peak separation. (**e**) Schematic of the excitation and emission processes of (VO_4_)^3−^ tetrahedrons in vanadate phosphor.

**Figure 3 molecules-25-00542-f003:**
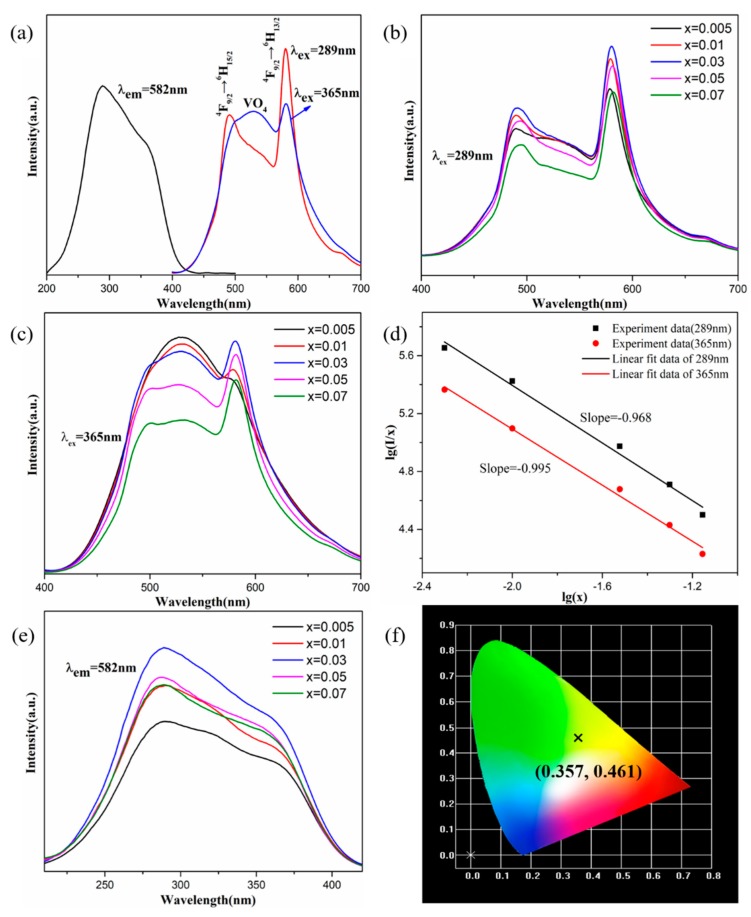
(**a**) PLE and PL spectra of the Na_2_YMg_2_V_3_O_12_:0.03Dy^3+^ sample. Emission spectra at (**b**) *λ*_ex_ = 289 nm and (**c**) *λ*_ex_ = 365 nm for different concentrations of Dy^3+^ in Na_2_YMg_2_V_3_O_12_:*x*Dy^3+^. (**d**) Linear fitting data of lg(*I*/*x*) versus lg(*x*) for the Na_2_YMg_2_V_3_O_12_:*x*Dy^3+^ phosphors. (**e**) Excitation spectra (*λ*_em_ = 582 nm) of Na_2_YMg_2_V_3_O_12_:*x*Dy^3+^. (**f**) CIE chromaticity coordinates of the Na_2_YMg_2_V_3_O_12_:0.03Dy^3+^ sample.

**Figure 4 molecules-25-00542-f004:**
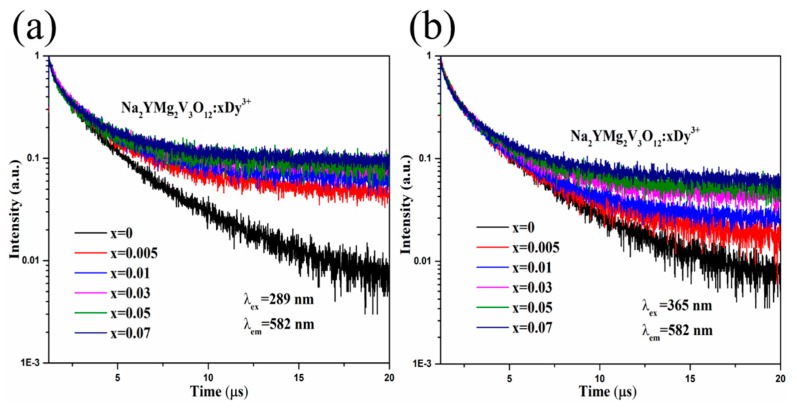
Decay curves of Na_2_YMg_2_V_3_O_12_:*x*Dy^3+^ with different concentrations of Dy^3^ excited at (**a**) *λ*_ex_ = 289 nm and (**b**) *λ*_ex_ = 365 nm, (*λ*_em_ = 582 nm).

**Figure 5 molecules-25-00542-f005:**
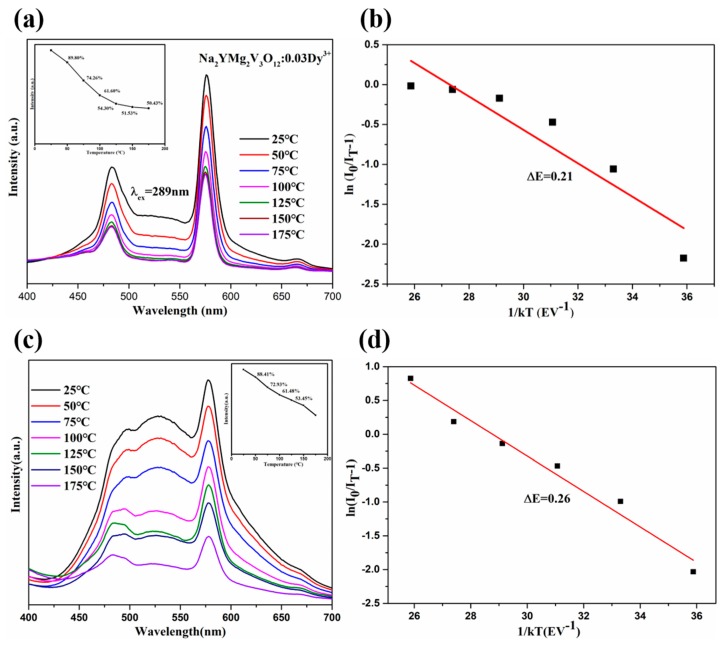
(**a**,**c**): PL spectra of the Na_2_YMg_2_V_3_O_12_:0.03Dy^3+^ phosphor at different temperatures (25–175 °C) excited at *λ*_ex_ = 289 and 365 nm, respectively. Insets show the PL intensities of Na_2_YMg_2_V_3_O_12_:0.03Dy^3+^ as functions of temperature. (**b**,**d**): Linear fitting curves of ln[*I*_0_/*I*(*T*) − 1] versus 1/*kT* for the Na_2_YMg_2_V_3_O_12_:0.03Dy^3+^ phosphor excited at 582 and 365 nm, respectively.

**Figure 6 molecules-25-00542-f006:**
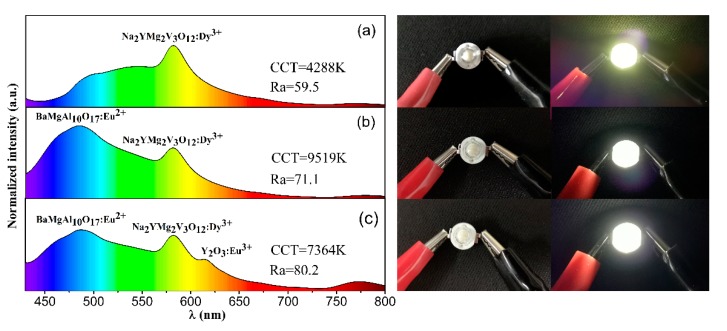
Electroluminescence (EL) spectra (left) and photographs (right) of (**a**) Na_2_YMg_2_V_3_O_12_:0.03Dy^3+^, (**b**) Na_2_YMg_2_V_3_O_12_:0.03Dy^3+^ with BaMgAl_10_O_17_:Eu^2+^, and (**c**) Na_2_YMg_2_V_3_O_12_:0.03Dy^3+^ with BaMgAl_10_O_17_:Eu^2+^ and Y_2_O_3_:Eu^3+^. The samples were incorporated into 365 nm InGaN LED chips with an injunction current.

**Figure 7 molecules-25-00542-f007:**
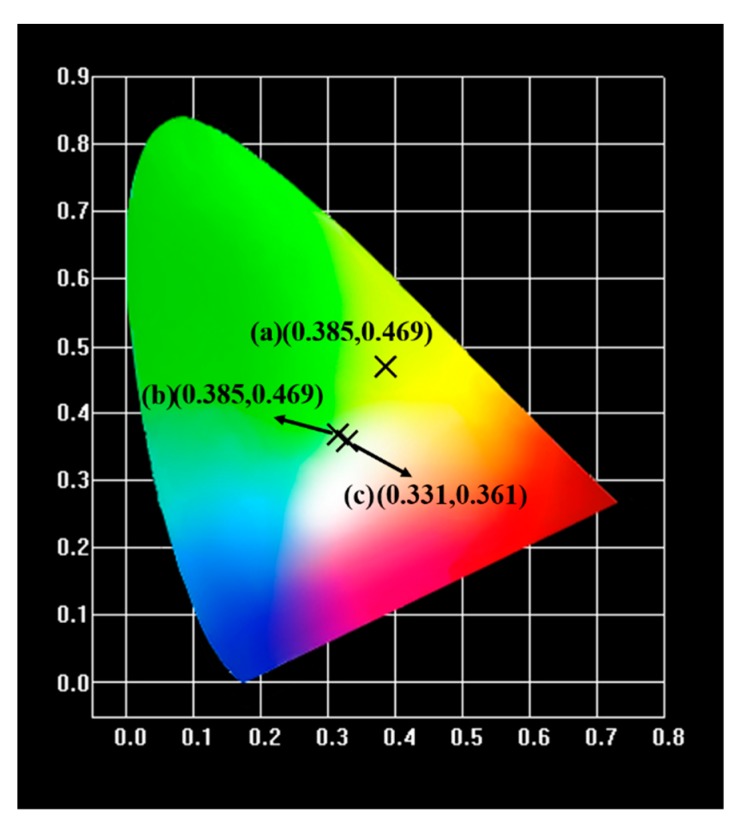
CIE coordinates of LEDs fabricated with (**a**) Na_2_YMg_2_V_3_O_12_:0.03Dy^3+^, (**b**) Na_2_YMg_2_V_3_O_12_:0.03Dy^3+^ and BaMgAl_10_O_17_:Eu^2+^, and (**c**) Na_2_YMg_2_V_3_O_12_:0.03Dy^3+^, BaMgAl_10_O_17_:Eu^2+^, and Y_2_O_3_:Eu^3+^ phosphors.

**Table 1 molecules-25-00542-t001:** Comparison of CIE chromaticity coordinates (*x*, *y*), correlated color temperature (CCT) (K), and lifetimes (μs) of Dy^3+^-doped phosphors.

Sample	(x, y)	CCT	Lifetimes	Reference
Na_2_YMg_2_V_3_O_12_: Dy^3+^	(0.357, 0.461)	4288	1.50	Present work
Sr_3_Y_2_(BO_3_)_4_: Dy^3+^	(0.300, 0.314)	5896	-	[42]
KBaY(MoO4)_3_: Dy^3+^	(0.431, 0.457)	3988	0.125	[43]
Na_3_Gd(VO_4_)_2_:Dy^3+^	(0.664, 0.335)	-	0.234	[22]
Ca_3_TeO_6_:Dy^3+^	(0.417, 0.460)	3730	0.506	[35]
NaLa(PO_3_)_4_: Dy^3+^	(0.292, 0.336)	-	0.78	[44]
NaCaPO_4_:Dy^3+^	(0.32, 0.37)	5962	0.604	[45]

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
