# Peer review of "Yellow Emission Obtained by Combination of Broadband Emission and Multi-Peak Emission in Garnet Structure Na_2_YMg_2_V_3_O_12_: Dy^3+^ Phosphor"

_molecules, 2020, doi:10.3390/molecules25030542_

Round 1

Reviewer 1 Report

Having examined your manuscript entitled “Yellow emission obtained by combination of 2 broadband emission and multi-peak emission in 3 garnet structure Na2YMg2V3O12: Dy3+ phosphor I note that you have made some interesting measurements related to the study of charge transfer in (VO4)3- groups in Na2YMg2V3O12 doped with different concentrations of  Dy3+   but before their publication several issues should be addressed. Therefore, it requires a mayor revision. Please check carefully.

Abstract, introduction and Conclusions

The author should highlight the relevance to develop Na2YMg2V3O12 based materials. It is not clear in the introduction why this matrix was selected and if the matrix has been studied previously. The authors should underline the innovative process, what is the advantage of the synthesis selected. Please give examples and compare with the state of the art.

Materials and Methods

What is the advantage and novelty of the synthesis process employed?

What is the particle size of the precursors selected?

Results and Discussion section:

The authors said that “The average particle size of the prepared phosphor is similar to the commercial phosphors”. Please include the particle size distribution of the powder prepared and the commercial powders.

From the SEM characterization, it seems that Na2YMg2V3O12:0.03Dy3+ powders start to sinter. It means the particle size could be higher. How this factor can have influence in the dispersion of the particles in the silica gel?

The authors don’t show any cross section of the InGaN chip after the coating with the powders embedded in the silica gel. Please add any micrograph.

What is the requirements of the silica gel, refractive index, thickness? The authors can check this details in the following paper and add as a reference,

Up-conversion emission of aluminosilicate and titania films doped with Er3+/Yb3+ by ion implantation and sol-gel solution doping, Surface and Coatings Technology, Volume 355, 15 December 2018, Pages 162-168, https://doi.org/10.1016/j.surfcoat.2018.01.056

The authors done a de-convolution of the emission band located at 524 nm resolving into two peaks by the Gaussian peak separation. The authors should take in account that is incorrect use of band de-convolution on the wavelength representation. Please read this article:

Potential problems in collection and data processing of luminescence signals, Journal of Luminescence, Volume 142, October 2013, Pages 202-211, https://doi.org/10.1016/j.jlumin.2013.03.052

The authors should provide quantitative values regarding other vanadate compounds to judge the values presented.

Unfortunately, the materials and their properties are not compared with existing state-of-the-art compounds or standards making it difficult to judge what improvements they offer. Specifically, all the photoluminescence studies present data in arbitrary units and there is no quantitative comparison with the luminescence properties of commercial, or other synthesized materials. The reader therefore cannot adequately judge the merits of your synthetic process. Therefore your article will not be considered, if you don’t include a reference phosphor powder.

Lakshmi Devi, C.K. Jayasankar,Spectroscopic investigations on high efficiency deep red-emitting Ca2SiO4:Eu3+ phosphors synthesized from agricultural waste,Ceramics International,Volume 44, Issue 12,2018,Pages 14063-14069, https://doi.org/10.1016/j.ceramint.2018.05.003.

Nakano, K. Kamimoto, N. Yokoyama, K. Fukuda, The effect of heat treatment on the emission color of p-doped Ca2SiO4 phosphor, Materials (Basel). 10 (2017). doi:10.3390/ma10091000.

Previously, a comparative study has been done by other authors as:

[R.E. Rojas-Hernandez, F. Rubio-Marcos, A. Serrano, E. Salas, I. Hussainova, J.F. Fernandez, Towards blue long-lasting luminescence of Eu/Nd-doped calcium-aluminate nanostructured platelets via the molten salt route, Nanomaterials. 9 (2019). doi:10.3390/nano9101473.

The authors discuss vaguely the applicability of this kind of compounds, and the requirements particle size, morphology, stability needed.

Therefore your article will not be considered, if the author does not include and add the changes proposed

Reviewer 2 Report

This manuscript presents interesting and important original results on a potential phosphor for solid-state lighting.
It is recommended that this manuscript is published in Molecules. However, before publication, extensive corrections are required, as detailed below.

The manuscript requires a great deal of language editing. There are numerous grammatical and structural mistakes throughout the manuscript. Professional editing is highly recommended.

Excitation for visible light emission at 289 nm and even 365 nm is totally impractical because LEDs emitting at such wavelengths are very expensive and emit little light when compared with visible blue LEDs.
Why do the authors think that, given economic and practical considerations, LEDs constructed as described in their manuscript can be practical commercial devices?

Line 69, 70: 'It is can be noted that NaHCO3 (>5 mol%) and Mg(OH)2 were used to compensate the volatilization loss.'
What does this mean?

Line 74-76: 'The as-prepared Na2YMg2V3O12:0.03Dy3+ phosphor was used to fabricated LED devices, commercial Y2O3:Eu3+ red phosphors, BaMgAl10O17:Eu2+ blue phosphors and InGaN chip with
dominating emitting at 365 nm (Shenzhen looking long technology Co., China).' Very badly written sentence that does not make any sense.

Line 76 and 77: 'The proper amounts of phosphors were added into the organic silica gel and mixed for 30 min.' What amounts were 'proper amounts'? What is organic silica gel? Is it silicone polymer?

Line 85: 'under 400V of operating voltage' What was this voltage applied to?

Line 213-215: Ra of 80 is very low for a multi-component 'balanced' phosphor. Can the authors comment on this?

Inset to figure 5(c) has mis-labelled horizontal axis ('wavelength', instead of 'temperature').

--------------------------------------------------- END ----------------------------

Round 2

Reviewer 1 Report

Thanks for the prompt reply and the modifications done.

I have only missed the following reply, that I did not find in the corrected version. Please add this details in the manuscript.

(7) What is the requirements of the silica gel, refractive index, thickness? The authors can check this details in the following paper and add as a reference,

Up-conversion emission of aluminosilicate and titania films doped with Er3+/Yb3+ by ion implantation and sol-gel solution doping, Surface and Coatings Technology, Volume 355, 15 December 2018, Pages 162-168, https://doi.org/10.1016/j.surfcoat.2018.01.056

Responses: Thanks very much for the reviewer’s kind suggestion. By referring to the above paper, the silica gel used to package LED chips need excellent light transmittance. [2] The light transmittance of the silica gel we chosen is 96% (thickness of 1mm). As can be seen from the paper, the transmittance will enhance with smaller thickness. In our research, the thickness of the silica gel is about 0.5mm, which can ensure good light transmission of the LEDs. We have enriched in the revised manuscript and cited the following paper as a reference.

Author Response

Response:Thanks very much for your suggestion concerning improvement to this paper. We have enriched in detail about this comment in the revised manuscript and cited the paper as a reference.

Reviewer 2 Report

Following suggestions from the reviewer, the authors have improved their manuscript. However, the English language use still requires some improvements. My suggestion is that the manuscript is given a further careful round of English editing to improve its language quality.

While I am mostly satisfied with responses to the majority of comments, the authors have not commented on the second point that was raised about the suitability of short wavelength UV LEDs as sources for pumping white light LEDs. The authors have provided a response but have completely avoided answering the question, saying basically that they have done it because everyone else is doing likewise. I'll not hold the manuscript from publication on that ground because I agree, at least, that many researchers have published on short wavelength UV pump LED-based white light LEDs but this method is of no practical significance at all as it will never see commercial utilization until inexpensive and efficient UV LEDs become available.

My recommendation is to publish this manuscript after some English language editing.

Author Response

Thanks very much for your suggestion concerning improvement to this paper. The revised manuscript has given a professional English editing to improve the language quality, and the certificate has been uploaded as an attachment.
